# Health Expenditure Data, Analysis and Policy Relevance in Australia, 1967 to 2020

**DOI:** 10.3390/ijerph19042143

**Published:** 2022-02-14

**Authors:** John R. Goss

**Affiliations:** Health Research Institute, Faculty of Health, University of Canberra, Canberra 2617, Australia; john.goss@canberra.edu.au

**Keywords:** health expenditure, health expenditure projections, disease expenditure, health expenditure policy

## Abstract

Since 1985, the Australian Institute of Health and Welfare (AIHW) has published 85 health expenditure publications. It has gradually extended the scope of these publications by extending the health accounts to detail expenditure by disease and age/sex, by State, Territory and remoteness and by Indigenous status. These enhanced health expenditure databases were then used to understand in detail the drivers of health expenditure. Understanding the drivers of health expenditure enables policy makers to understand where to intervene so as to maximise the health improvements that arise from health expenditure growth.

## 1. History of Health Expenditure Data Collection and Publication in Australia

The first comprehensive set of health expenditure numbers for Australia was published by John Deeble in 1967 [1]. These estimates set the scene for the development work by Deeble and Scotton, leading to the introduction of universal health insurance in Australia in 1975 [2].

The first health expenditure publication by the Commonwealth Department of Health “Australian health expenditure 1974–75 to 1977–78: An analysis” was issued in 1980 [3]. This was followed by 3 updates in 1981, 1983 and 1985 [4,5,6].

The health expenditure collection and publication function was transferred to the Australian Institute of Health (AIH) when it was established as a separate Division within the Commonwealth Department of Health in 1985. The first health expenditure monograph published by the AIH was published in 1988 [7].

The author of this paper first became involved in health expenditure collection, analysis and publication at the AIH in 1986 and concluded his work on health expenditure at the Institute in 2010, so this paper reflects the views of an insider. The perspective of the author needs to be understood in interpreting the views expressed in this paper.

Since then, the AIH, or the Australian Institute of Health and Welfare (AIHW) as it became, has published 85 health expenditure publications [8]. Dissemination of information is a prime part of the mission of the AIHW, whereas the prime role of the Commonwealth Department of Health is policy advice and implementation and these functions tend to crowd out its information dissemination role. Additionally, because the Institute became an independent statutory authority of the Commonwealth Government in 1987, the analysis that accompanies the data is bolder and more independent than analyses from government health departments.

The Australian health expenditure accounts are mostly collated according to the rules and classifications of the System of Health Accounts [9]. Such classification systems have their inadequacies in that they have to classify expenditure into one category or another, so the multipurpose nature of most expenditure is not captured. However, these classification systems have the advantage that they classify expenditure fairly consistently across countries and across States and Territories within Australia.

Having a consistent set of definitions for health expenditure in Australia’s Metadata Online Repository (METeOR) is a necessary first step in developing datasets which are consistent across jurisdictions, but much work needs to be carried out to encourage the Australian States and Territories and the Commonwealth Government to provide data according to these definitions.

## 2. Aboriginal and Torres Strait Islander Health Expenditure

Australia was the first country to comprehensively estimate how much was spent on health services for its Indigenous Aboriginal and Torres Strait Islander population. This was carried out in a report by Deeble, Mathers, Smith and Goss published in 1998 [10]. This report estimated that per person health expenditure from all sources for Indigenous people in 1995–1996 was 8% higher than for non-Indigenous people. As the health status of Indigenous people was so much worse, with a life expectancy gap of at least 12 years, it was clear that the 8% higher per person expenditure was not enough to address the much greater need for health services of Aboriginal and Torres Strait Islander people. Up until this publication, it was the popular view that large amounts were spent on health services for Indigenous people and a significant portion of this expenditure was wasted. After this publication, that view was no longer tenable. Spending was shown to be particularly low for Australian Government-funded medical benefits and pharmaceuticals. This report led the Commonwealth Department of Health to increase substantially its funding for Aboriginal and Torres Strait Islander health services both to start to address the identified deficit in funding of Indigenous health and because the report showed the States were funding a greater proportion of Indigenous health expenditure as compared to the Commonwealth Government.

The report also exposed deficiencies in the identification of Aboriginal and Strait Islander people in health data collections and helped lead to improvements in identification in these collections.

Indigenous health expenditure estimates have continued to be refined over the years [11], including estimates being made of Indigenous health expenditure by remoteness and by disease [12]. The ratio of Indigenous to non-Indigenous health funding per person has increased substantially from 1.08 in 1995–96 to 1.30 in 2015–16 [13].

### 2.1. Disease Expenditure Data

The first comprehensive disease expenditure data for Australia were published by the AIHW in 1998 for the reference year 1993–94 [14]. These were world leading data. Many countries had published data for expenditure for particular diseases, and particularly for the government-funded portion of that expenditure. However, no other country had published expenditure data for each disease, for government- and private-funded expenditure and for almost all areas of expenditure. This disease expenditure publications dissected expenditure by disease for 90% of recurrent expenditure in 1993–94. Disease expenditure data were published subsequently for 2000–01, 2004–05, 2015–16 and 2018–19 [15,16,17,18].

In 2018–19, musculoskeletal disorders accounted for 10.3% of recurrent health expenditure which could be allocated by disease, followed by cardiovascular disease at 8.7%. Cancer accounted for 8.6% of expenditure, mental illness 7.7% and injury 7.6%. Reproductive and maternal conditions accounted for 6.7% and oral disorders for 6.5%. It is noteworthy that while much of the discourse in health is about interventions that reduce mortality, that leading reasons for health expenditure include assisting mothers to give birth and the low mortality conditions of musculoskeletal disorders, mental illness and oral disorders.

The great strength of the disease expenditure analyses performed by the AIHW is that it is performed within the standard health expenditure framework. Many disease costing studies attempt to estimate the total social cost of a disease including indirect costs such as the loss of productivity due to a person dying from disease. The problem with this approach is that when the costs of all the different diseases are added up, the total number is many times total health expenditure. The AIHW approach allows the expenditure caused by a particular disease to be compared to actual real world health expenditure.

### 2.2. Public Health Expenditure

A detailed dissection of public health expenditure into 9 categories for each of the States and Territories and for the Commonwealth Government was produced in 2001 for the reference year 1998–99 [19]. These detailed data were published until the reference year 2008–09 [20], after which it ceased as the Commonwealth Department of Health stopped funding it. The nine categories of expenditure were immunisation (28% of public health expenditure in 2008–09), health promotion (19%), communicable disease control (12%), food standards and hygiene (2%), breast cancer and cervical screening (15%), and prevention of harmful drug use and public health research (7%). The per person public health expenditure was similar across the 6 States, but was 50% higher for the Australian Capital Territory) (ACT) and 300% higher for the Northern Territory (NT) in 2008–09. Over the 10 years for which these detailed data were published, the variation of State per person expenditure from the national mean State per person expenditure reduced by 50%.

Public health expenditure, as recorded in these reports, was $1014 million in 2000–01 [21]. In addition to this core public health expenditure, there was substantial expenditure on primary health care services and pharmaceuticals which reduced hypertension and cholesterol. This public health-related expenditure was $2140 million in 2000–01.

In total, core public health and public health-related expenditure came to $3154 million in 2000–01. Although this was only 5.9% of total recurrent health expenditure in 2000–01, it was responsible for a disproportionate proportion of the improvement in health that occurred around this period. So, for example, reductions in smoking, systolic blood pressure and cholesterol accounted for 74% of the male decline and 81% of the female decline in the coronary heart disease mortality rate in the period 1968–2000 [22].

## 3. Drivers of Health Expenditure Growth

Health expenditure grows every year, and usually at rates which are higher than other sectors in the economy. Health expenditure as a percentage of GDP has grown from 7.6% of GDP in 1978–79 to 10.0% of GDP in 2018–19 [23]. The question as to why health expenditure is growing at such a high rate is frequently asked. Following on from this, questions are asked as to whether the growth in health expenditure is sustainable and whether this increase in expenditure is achieving value for money.

To answer these questions, we must first understand the drivers of health expenditure. How much of health expenditure growth is due to the demographic factors of ageing and population growth? How much health expenditure growth is due to changing disease and risk factor levels? How much growth is due to the higher price of health goods and services relative to prices in the rest of the economy? Additionally, how much of health expenditure growth is due to higher rates of services provided per case of disease? Analysis of the drivers of health expenditure growth 50 years ago was very much in its infancy but as the years have gone by, decomposition analysis of health expenditure drivers has become more complete and sophisticated.

Table 1 below shows the decomposition of growth in 3 major areas of health expenditure for from 2000–01 to 2011–12 and from 2011–12 to 2018–19. The decomposition uses the Das Gupta decomposition method [24,25]. These three areas of hospital admitted patient services, medical services and pharmaceuticals together accounted for 55% of recurrent health expenditure in 2018–19. Health expenditure for these three areas grew in real terms at an average pace of 5.0% per year from 2000–01 to 2011–12 and at an average pace of 3.1% per year from 2011–12 to 2018–19. (Expenditure is calculated in real terms by deflating expenditure by the Gross National Expenditure (GNE) deflator. The GNE deflator is a good measure of general inflation in the economy as a whole and is the most appropriate deflator to use when comparing the value of money spent in the health sector as compared to money spent elsewhere.)

The 5.0% annual growth from 2000–01 to 2011–12 can then be decomposed into the demographic component of 1.8% per year and the non-demographic component of 3.1% per year. The demographic component is then decomposed into the population growth component of 1.1% per year and the ageing component of 0.7%. The non-demographic component can be decomposed into three factors—excess health price inflation which adds 0.34% per year to real health expenditure growth and changing rates of disease which reduces health expenditure growth by 0.08% per year. (The projections section discusses more about this surprising result.) The residual component of health expenditure growth adds 2.8% per year. This component represents how much real health expenditure has increased due to more health goods and services being delivered per case of disease.

This increase in services per case of disease is the key parameter in determining whether an increase in health expenditure is value for money. One would normally expect to see an increase in health system attributable outcomes of at least 2.8% per year in order to justify an increase in services per case of disease of 2.8% per year. From 2003 to 2011, a measure of health outcomes—age-standardised Disability Adjusted Life Year (DALY) rates—declined by 1.1% per year. This is a strong indication that the rate of increase in health expenditure in this period was not value for money.

Further work needs to be carried out to ascertain whether there really was a decline in health productivity in this period, but disease expenditure data enable analysis to be performed as to whether increases in disease expenditure inputs result in commensurate disease improvements.

There was a significantly lower growth rate in real expenditure from 2011–12 to 2018–19 as compared to from 2000–01 to 2011–12 of only 3.1% per year.

Almost all of this lower growth is due to services delivered per case of disease growing at 1.2% per year as compared to the 2.8% per year rate of growth for this factor from 2000–01 to 2011–12.

In order to ensure that health expenditure grows at an optimal rate, it is primarily the growth in services per case of disease which must be controlled. This factor grows due to changes in treatment practices, changes in technology and changes in consumer preferences.

Some of the systems which control health expenditure growth, such as hospital casemix funding, have been unhelpful as they have allocated resources without understanding which growth is necessary and improves health, and which growth is wasteful and detracts from health. The Pharmaceutical Benefits Advisory Committee and the Medical Services Advisory Committee have followed a better way of restraining wasteful expenditure by evaluating whether new pharmaceuticals or new medical services are cost-effective.

Understanding growth in services per case of disease, and how this growth results in health outcome improvements, is another approach to fostering increases in expenditure which improve health.

## 4. Health Price Increases and General Inflation

Health prices generally increase faster than general inflation because the health sector is dominated by services, and the price of services in a growing economy goes up on average faster than the price of goods [26,27]. This amount by which health prices increase faster than general inflation is called ‘excess health price inflation’ and, as shown above, is a significant driver of health expenditure increases.

However, it is important to understand that excess health inflation is different for each health price index, and the extent of excess health inflation varies over time.

Table 2 shows excess health price inflation relative to the GNE deflator for health prices as a whole, and for hospital, medical, dental and pharmaceutical prices.

From 2002–03 to 2009–10, excess health price inflation for health prices as a whole was 0.72% per year. Excess hospital price inflation was 0.99% per year and excess medical price inflation was 1.51% per year. (The medical price deflator used here is the Medicare medical service fee charged deflator.) The Pharmaceutical Benefits Scheme (PBS) recorded a negative pharmaceutical excess price inflation of 1.63% per year during this period.

From 2009–10 to 2014–15, overall excess health price inflation was unusually negative at −0.32% per year. Although excess hospital price inflation was positive at 0.37% per year, excess medical, dental and pharmaceutical price inflation were all negative, leading to the overall negative result.

From 2009–10 to 2018–19, excess medical price inflation was almost always negative due to the government severely limiting increases in the benefits proscribed by the Medicare Benefits Schedule (MBS). The 11% annual average increase in excess pharmaceutical price inflation from 2014–15 to 2016–17 was due almost entirely to expensive Hepatitis C pharmaceuticals being added to the PBS.

## 5. Health Expenditure Projections

Projections for components of Australian health expenditure have been undertaken for many years in Australia, e.g., the Commonwealth Intergenerational Reports project Commonwealth health expenditure 40 years into the future [28]. However, more sophisticated projections only became possible when burden of disease analyses became available. Burden of disease analyses estimate the overall impact of disease by estimating the impact of disease in reducing life expectancy and its impact in increasing illness and reducing functioning. The overall burden of disease is measured using a metric called the Disability Adjusted Life Year (DALY). The DALY consists of the premature mortality component called the Years of Life Lost (Years of Life Lost), and healthy life years lost due to illness and reduced functioning called the Years of Life lost due to Disability (YLD). The burden of disease analyses estimate not just the burden imposed by each disease, but also the prevalence, incidence, severity and sequelae of disease, and the risk factors that increase the risk of disease [29].

The first projection of Australian health expenditure that took into account disease projections was a report for the United Nations World Economic and Social Survey 2007 by Vos, Goss, Begg and Mann [30].

Then, in 2009, Goss reworked this projection for the National Health and Hospitals Reform Commission [31]. This study produced some surprising results.

First, this study estimated that changing disease rates over the 30 years 2002–03 to 2032–33 were expected to lead to a net reduction in health expenditure of $2.3 billion. Although expected increases in the disease rates for diabetes and other diseases would lead to increased expenditure of $4.7 billion, expected decreases in expenditure on heart disease, cancer and other diseases would lead to savings of $7.4 billion, leading to net savings of $2.3 billion.

Second, ageing was not the main driver of health expenditure that many people expected. Of the projected increase in health and residential aged care expenditure of $161 billion, only $38 billion (23%) was due to ageing.

Third, the biggest factor expected to drive health expenditure increases was the growth in the amount of health services provided per case of disease. This factor was expected to grow by $81 billion in the 20 years to 2032–2033, which was 50% of the overall increase. The growth of this factor is mostly under the control of the health system (in contrast to the other drivers of health expenditure which are mostly not). For each case of presenting disease, providers mostly have the power to choose over time to provide more (or less) services per case of disease, and consumers also have some power to demand more (or less) services per case of disease.

This projection model was used to inform recommendations to the Commonwealth Government by the National Health and Hospitals Reform Commission in areas such as the impact of a reduction in smoking, an increase in aged care places, the improved treatment of diabetes and a reduced rate of increase in obesity rates [32].

## 6. Conclusions: Impact of Health Expenditure Data and Analysis on Health Policy in Australia, 1967 to 2020

Health expenditure data have been influential in shaping debates about health policy in Australia and in shaping health policy itself.

The health expenditure data that John Deeble collected and analysed in the 1960s were critical in shaping the policy recommendations from Deeble and Scotton that were crucial in the establishment of Medibank in 1975.

Information about what was actually spent on health services for Aboriginal and Torres Strait Islander people changed the policy debate from one focussed on reducing ‘waste’ in spending on health services for Aboriginal and Torres Strait Islander people to addressing major unmet needs in expenditure on these services.

For decades, health expenditure as a proportion of GDP has been a marker of the debate as to whether too much or too little was being spent on health services in Australia. However, only when there is a detailed understanding of the drivers of health expenditure as a proportion of GDP is it possible to have an informed debate as to how much should be spent on health and where it should be spent. This understanding of what drives health expenditure depends on a detailed understanding of where the money is spent—how much is spent by hospitals, medical practices, pharmacies, etc., how much is spent by governments, health insurance funds and individuals, how much is spent for each age/sex group, how much is spent for the prevention and treatment of each disease, how much is spent for different socioeconomic groups, for Aboriginal and Torres Strait Islanders and for people living in different regions of Australia. In the last 50 years, we have developed our understanding of the details of what is spent on health and for whom and for what purpose, so that now, when we link this detailed expenditure information to the health outcomes it engenders, we are able to more wisely allocate our health expenditure so as to achieve higher-quality health care for all.

## Figures and Tables

**Table 1 ijerph-19-02143-t001:** Drivers of real health expenditure growth, from 2000–01 to 2011–12 and from 2011–12 to 2017–18.

Drivers of Real Health Expenditure Growth	2000–01 to 2011–12	2011–12 to 2017–18
** *Total real annual average growth* **	** *5.0%* **	** *3.1%* **
**Demographic growth**	**1.8%**	**1.7% ^1^**
Population growth	1.1%	1.1%
Ageing	0.7	0.6
**Non-demographic growth**	**3.1%**	**1.4% ^2^**
Excess health price inflation	0.34%	0.14%
Disease rate changes	−0.08%	0.07%
Growth in services per case of disease	2.8%	1.2%

Numbers in Table 1 calculated by author from [23]. ^1^ “Demographic growth” combines “Population growth” and “Ageing” ^2^ “Non-demographic growth” combines “Excess health price inflation”, “Disease rate changes” and “Growth in services per case of disease”.

**Table 2 ijerph-19-02143-t002:** Excess health price inflation relative to GNE deflator, annual average growth, from 2002–03 to 2018–19.

Excess Health Price Inflation	2002–03 to 2009–10	2009–10 to 2014–15	2014–15 to 2016–17	2016–17 to 2018–19	2002–03 to 2018–19
Total excess health price inflation	0.72%	−0.32%	0.86%	0.05%	0.33%
Excess hospital price inflation	0.99%	0.37%	0.63%	1.05%	0.76%
Excess medical price inflation	1.51%	−0.32%	−0.59%	−0.72%	0.39%
Excess dental price inflation	1.76%	−0.95%	−2.04%	−0.92%	0.09%
Excess pharmaceutical price inflation	−1.63%	−4.32%	11.28%	−5.29%	−1.43%

Numbers in Table 2 calculated by author from [23].

## Data Availability

All data used are referenced in References.

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
