# Peer review of "Health Expenditure Data, Analysis and Policy Relevance in Australia, 1967 to 2020"

_ijerph, 2022, doi:10.3390/ijerph19042143_

Round 1

Reviewer 1 Report

This is an excellent review which, as promised in the title, highlights the policy relevance of the health expenditure publications in Australia.

The main weaknesses is what is not included, particularly any identification of the weaknesses of the publication series in particular:

  • the use of the national accounts framework means that expenditure on health professional education in universities and colleges is not included in the data, so some education costs are in the series (those borne by hospitals and other health services) and some are not)
  • to what extent is there consistency between states in reporting? E.g. of administrative costs and, perhaps, public health expenditure

It might also be worth exploring (not necessary) whether the analyses of cost growth in the health expenditure studies should be accompanied by analyses of productivity growth an issue beginning to be addressed in England (Bojke et al. 2017).

Bojke, Chris, Adriana Castelli, Katja Grašič, and Andrew Street. 2017. "Productivity Growth in the English National Health Service from 1998/1999 to 2013/2014." Health Economics 26 (5): 547-565. https://doi.org/10.1002/hec.3338.

Reviewer 2 Report

General Comments

This is an interesting article, reviewing the health expenditure data (and its historical development) in Australia.  The point that this data allows health expenditure to be examined – rather than just the social cost of disease – is an important distinction and allows for some useful analysis.  While the analysis is useful – and necessarily high-level – some more detail and/or discussion in places would be useful.

Specific Comments

  • Page 1, Lines 25-36: In the first of these two paragraphs, the author states that it was established as a division of the DoH, but in the second of these paragraphs the author states that the Institute is an independent statutory authority. Has the AIHW been separated out of the Department of Health?  If so, then this should be stated explicitly.
  • Page 2, Lines 90-91: Why were these data discontinued after the year 2008-09?
  • Page 3, Lines 131-133: Do the figures differ significantly if alternative deflators are used?
  • Page 3, Lines 134-143: More details on the methodology for this decomposition would be useful.
  • Page 4, Lines 146-147: Have health outcomes increased by 2.8% per annum? If so, how are these outcomes measured?
  • Page 4, Lines 153-154: There appears to be a reference to a footnote in Table 1, but no footnote (or endnote) appears in the article.
  • Page 4, Lines 160-162: This is quite a strong statement. Has an analysis been undertaken of these attempts to gauge their efficacy?
  • Page 4, Lines 170-172: A reference to back up this statement this would be useful.
  • Page 5, Lines 195-197: Was this entirely down to a single drug? While expensive drugs have the ability to inflate health expenditures, given the relatively low number of people suffering with Hepatitis C, this seems to be an extraordinary level of excess pharmaceutical price inflation for just one drug.
  • Page 5, Lines 217-218: Should this be 30 years or should the second year be 2022-23?
  • Page 6, Lines 230-232: In theory this may hold, but in practice, providers may not be able to exercise full choice, particularly in relation to providing less services per case, while consumers may not have sufficient information to demand more.
  • Page 6, Lines 252-258: Some of these areas have not been explored in this article, but have led to significant debate, particularly in relation to public versus private healthcare. Perhaps some high-level mention of these factors would be useful.

Minor Comments

  • Page 2, Line 59: This should read health funding per person.
  • Page 4, Line 150: One almost should be deleted.
  • Page 4, Line 151: The fro should either be for or from.

Reviewer 3 Report

I really liked reading this paper. I learned a lot about the nature of health care expenditure in Australia over time. I have a few minor questions and comments about the paper.

  1. I wonder what is driving the disparity in health outcomes for indigenous people in Australia compared to non-indigenous population. Lower public spending is one explanation. However, are there differences in behavior or circumstances that simultaneously put them at a higher risk? This might be worth investigating further.
  2. Lines 73 to 76 states that "it is noteworthy that assisting mothers to give birth and low mortality conditions of musculoskeletal disorders, mental illness, and oral disorders are leading reasons for health expenditure". I do not understand why it is noteworthy. I also see that the percentage of expenditure is bigger for cardiovascular diseases (8.7%) and cancer (8.6) than mental illness (7.7%), reproductive and maternal conditions (6.7%), and oral disorders (6.5%). 
  3. What is the reason behind the 50% reduction in variation between State per person expenditure from the national mean State per person expenditure?
  4. Line 150 has an extra "almost" that needs to be deleted. 

Reviewer 4 Report

It is very difficult to assess this paper. The content is interesting and the presentation is good. However, I have strong doubts that this paper is appropriate for publication in IJERPH. It is purely descriptive. There is no methodology but just reporting what was done. It reads more like a story from personal experiences than a scientific paper. The selection of papers and documents included cannot be supported by any kind of systematic review.

  • title: the title does not fit to the paper. Why 1970? Don't you start with 1985?
  • Drivers of health expenditure: you need a methodology to determine drivers.
  • Non-Australian readers would need some more background on the Australian health care system (and partly on Australia. For instance, not everybody knows what Torres Strait is).
  • the text includes many judging phrases, such as "pathbreaking", "only 8 %" ...
  • The section on regional expenditure is almost neglected. Here I would expect  a map and detailed explanation.
  • we would definitely need a methods section. How did you select the papers and documents?
  • the conclusions are full of statements that are not based on any scientific methodology. How did you draw these conclusions based on the documents?

in principle, this information is relevant and intesting as a starting point for a real methods-based analysis. But as it is now I would not support publishing.

Round 2

Reviewer 4 Report

The authors have addressed some issues, but my main concern is still the same: it is not a scientific paper based on sound methodology and with unique results. It is more or less a starting point for other research.
